# Direct sequencing of *Leishmania donovani* from patients in Garissa County, Northern Kenya, reveals a newly emerging intra-specific hybrid genotype

Vane Kwamboka Omwenga[1,2], Gathii Kimita[1], Damaris Matoke-Muhia[3], Cyrus Ayieko[2], Clement Masakhwe[1], Mohamed Hussein Ibrahim[4], Senne Heeren[5,6], Jean-Claude Dujardin[5], Pieter Monsieurs[7], Malgorzata Anna Domagalska[8]*, John Waitumbi[1]*

**1** Kenya Medical Research Institute/Walter Reed Army Institute of Research – Africa, Basic Science Laboratory, Kisumu, Kenya, **2** Department of Zoology, Maseno University School of Biological and Physical Science, Kisumu, Kenya, **3** Kenya Medical Research Institute, Centre for Biotechnology Research Development (CBRD), Nairobi, Kenya, **4** Department of Health, Garissa County Referral Hospital, Garissa County Kenya, **5** Institute of Tropical Medicine, Molecular Parasitology Unit, Antwerp, Belgium, **6** Department of Microbiology, Immunology and Transplantation, Rega Institute for Medical Research, Katholieke Universiteit Leuven, Leuven, Belgium, **7** Institute of Tropical Medicine, Trypanosoma Unit, Antwerp, Belgium, **8** Institute of Tropical Medicine, Experimental Parasitology Unit, Antwerp, Belgium

* john.waitumbi@usamru-k.org (JW); mdomagalska@itg.be (MAD)

## Abstract

### Introduction

*Leishmania*sis is endemic in many countries, and in Kenya outbreaks of visceral leishmaniasis (VL) commonly occur in Garissa, Isiolo, Marsabit, Turkana, and Wajir. Despite the rising frequency of VL outbreaks, there is limited data on the genetic structure and epidemiology of *Leishmania* parasites from these regions. This study used molecular methods to characterize *Leishmania* parasites collected at Garissa County Referral Hospital during the 2019–2022 VL outbreak.

### Methods

286 blood samples, collected from patients suspected of having VL at Garissa County Referral Hospital between 2019 and 2022 were used. *Leishmania* parasites were screened at genus level by a quantitative real-time PCR assay targeting the arginine permease gene AAP3 (AAP3-qRT-PCR). Species identification and targeted gene sequencing were made on Illumina MiSeq using PCR amplicons of *Hsp70* gene and ITS regions. Whole genome sequencing (WGS) was performed directly on eight selected blood samples using a target enrichment method after which data was analyzed using phylogenomic tools.

**Data availability statement:** All sequencing data are available at https://www.ncbi.nlm.nih.gov/bioproject/PRJNA1251555 for whole genome sequences. Hsp70 and ITS loci are available under GenBank accession numbers PQ700286–PQ700371 and PV557379–PV557457, respectively. All other data are available in the main text or supplementary material.

**Funding:** This study was supported by the Armed Forces Health Surveillance Division and its Global Emerging Infections Surveillance and Research Branch (grant numbers ProMIS P0089_24_KY to JW). SH was supported by the Research Foundation Flanders (grants G092921N); MAD acknowledges support from the "Departement Economie, Wetenschap en Innovatie" (Department of Work, Economics, Science, Innovation & Social Economics of the Flemish Government, WEWIS). The funders had no role in study design, data collection and analysis, decision to publish, or preparation of the manuscript.

**Competing interests:** The authors have declared that no competing interests exist.

## Results

By AAP3-qRT-PCR, 128/286 (45%) blood specimens were determined to have *Leishmania* parasites. We obtained 86 *Hsp70* and 79 ITS sequences that phylogenetically clustered with the *L. donovani* species complex. By WGS, the eight selected samples had *L. donovani* s.s., and clustered in two separate groups: one similar to the previously reported *L. donovani* group 5 and the other constituted a new and intra-specific hybrid genetic variant not reported previously. In all the 8 Kenya samples, we found SNPs in genes previously shown to be involved in *L. donovani* resistance to Antimony, Amphotericin B and Miltefosine.

## Conclusion

This pilot study reveals the complex nature of *Leishmania* genetic structure in Kenya and sheds light on the genomic polymorphism of *L. donovani* in this region, which in turn, may explain the evolving threat of VL in the region. As caveat, the genomic signatures of drug resistance genes that were identified should be interpreted with caution until their functional implication is clarified in future phenotypic studies.

### Author summary

Visceral leishmaniasis (VL), or kala-azar, is a deadly disease if untreated. In Kenya, outbreaks of the disease commonly occur in Garissa, Isiolo, Marsabit, Turkana, and Wajir. During the 2019–2022 outbreak in Garissa, we collected samples from patients reporting at Garissa County Referral Hospital to study the genetic structure of the parasites' population. To avoid biases from lab-grown parasites, we used advanced genetic methods on parasites directly obtained from patients' blood. We identified two parasite groups: one related to a previously known group 5 (Ldon5) from Kenya and Ethiopia, and a new group with mixed ancestry from Sudan, Ethiopia (Ldon3), and Iraq (Ldon4). Notably, we found mutations that might be linked to drug resistance. These findings are crucial because they help in understanding how the parasites evolve and spread. As caveat, the presence of drug resistance genes should be interpreted with caution until their functional implication are clarified in future studies.

## Introduction

Leishmaniasis is a life-threatening vector-borne tropical disease caused by protozoan parasites of the genus *Leishmania* and transmitted by female Phlebotomine sand flies [1]. The disease is endemic in 99 nations including Kenya (https://www.who.int/data/gho/data/themes/topics/gho-ntd-leishmaniasis), and is responsible for roughly 1 million cases and 50,000 fatalities annually [2]. The disease presents in three main forms influenced by the infecting species [3,4]: cutaneous leishmaniasis (CL) and

its subtypes; visceral leishmaniasis (VL, also known as Kala azar), [5]; and muco-cutaneous leishmaniasis (MCL) [6]. Of these, VL is the most serious and ranks as the second deadliest protozoan disease after malaria [7].

The distribution of leishmaniasis varies geographically with approximately 90% of global visceral cases occurring in Sub-Saharan Africa, Southeast Asia, and Brazil [7]. For a long time, the hotspot for VL was concentrated in the Indian sub-continent (ISC), but due to the success of recent elimination program in the ISC, East Africa now appears to have become a VL hotspot, where there is limited elimination efforts [8]. Some of the key factors that contributed to near elimination of VL in ISC included political commitment, targeted implementation that included treatments and follow-up of cases, strong surveillance among others [9,10]. In East Africa, leishmaniasis is widespread in Kenya, Ethiopia, Sudan, Uganda, Somalia and Eritrea [11]. Kenya specifically reports endemicity of the disease in the Rift Valley, Eastern and Northeastern regions, with outbreaks of VL occurring in sub-counties of Garissa, Isiolo, Marsabit, Turkana and Wajir [12], with approximately 5 million people being at risk of infection annually [13]. This underscores the urgency of conducting concerted epidemiological research to better understand and combat the disease in the country. The classification of *Leishmania* species remains contentious, with up to 53 species identified, 21 of which are known to infect humans [5,14]. Based on their geographical distribution, these species are grouped into New World -found in the Americas, and Old World distributed across Europe, Asia and Africa [15,16]. In Eastern Africa, VL is primarily attributed to two species of the *donovani* complex, *L. donovani* and *L. infantum*. Notably, the geographical distribution of these species can lead to genetic variations due to changes in virulence genes and potential hybrid formation, complicating our understanding of leishmaniasis [17–20].

In 2020, a phylogenetic classification of the two species of the *donovani* complex, based on Whole genome sequencing (WGS) analysis of 151 cultured isolates was published [21]. Accordingly, 5 groups were identified in *L. donovani* (Ldon1–5), 3 of them being endemic in Africa and Middle East and the two others in the Indian sub-continent. WGS offers a pan-genome perspective on parasite evolution with a high discriminatory power. This offers a unique reference frame for several applications like tracking the source of outbreaks, monitoring transmission patterns or identifying new genotypes [22].

Present study used targeted gene sequencing and untargeted WGS to directly characterize the *Leishmania* parasites collected from patients' blood at Garissa County Referral Hospital, in Northeastern Kenya during the 2019–2022 VL outbreak and situate them in the phylogenetic reference frame mentioned above. Noteworthy, we applied here for the first time in Africa, a method for direct genome sequencing of *L. donovani* in blood samples from VL patient, without the need for parasite isolation and culture [23,24]. Garissa and neighboring counties such as Kitui experience recurring outbreaks, probably because the semi-arid ecology provides ideal conditions for sand flies, particularly around termite mounds that offer microhabitats with moisture and organic matter for breeding [25]. Additionally, poverty, limited healthcare access, overcrowding and poor housing conditions especially in refugee camps like Dadaab, intensify disease spread [26].

## Materials and methods

### Ethics statement

This study was conducted as part of a public health surveillance activity following a request by the Ministry of Health, Garissa County to provide diagnostic and laboratory support during a protracted leishmaniasis outbreak in Garissa County. The use of these routine surveillance data in research is considered acceptable because of the public health impact, the fact that risks to the subjects are minimized through data deidentification, and the impracticability of seeking retrospective consent (ref to the WHO guideline). The MOH provided the samples for diagnostic support and we recorded the bio-specimens. The approval to provide diagnostic and laboratory support, including whole genome sequencing was approved by the Kenya Medical Research Institute Scientific and Ethics Review Unit (KEMRI SERU, # 4634) and the Walter Reed Army Institute of Research, Human Subject Protection Branch, (WRAIR #2699). Whole genome sequencing of eight samples (S1 Table) conducted at Institute of Tropical Medicine, Antwerp (ITM) using *Leishmania* target enrichment (SureSelect Sequencing, SuSL-seq) was approved by the ITM IRB (code 45/2024).

## Study site and design

The study utilized 286 archived blood samples collected from patients suspected of having visceral leishmaniasis during the 2019–2022 VL outbreak in Garissa and Kitui counties, Kenya. Patients' blood samples were collected at Garissa County referral hospital (S1 Fig). Consent was not obtained from the patients because this was a public health activity. The risks to the subjects were minimized through data deidentification. This study applied WGS to characterize *Leishmania* directly from patient blood samples, eliminating the need for parasite isolation and culture.

## DNA extraction and screening for Leishmania parasites

DNA was extracted from the archived blood samples using the MagMAX CORE, Nucleic Acid Purification reagents (Thermo Fisher Scientific, USA). For detection of *Leishmania* parasites, extracted DNA was screened at genus level by quantitative real-time PCR of the arginine permease gene AAP3 (AAP3-qRT-PCR). The following primer sets and probe were used [27]: forward - (AAP3), 5' GGC GGC GGT ATT ATC TCG AT 3', reverse - (AAP3), 5' ACC ACG AGG TAG ATG ACA GAC A 3' and probe - FAM 5' ATGTCGGGCATCATC 3' NFQ. All reactions were conducted in a 10.0µL reaction volume comprising 5.0µL of 2x SensiFAST master mix (Meridian Bioscience USA, https://www.meridianbioscience.com), 0.8µL of each primer at a concentration of 10 µM, 0.4µL of probe at concentration of 10µM, 2.0µL of DNA, and 1.0µL of PCR water (Thermo Fisher Scientific, USA). Each reaction included known positive controls of *Leishmania* spp (*L. donovani* and *L. major*). Non-target controls (PCR water) was used as negative control to track contamination.

## Species identification and targeted genotyping

DNA was used to PCR amplify the *Hsp70* gene and ITS region using primer sets as published elsewhere [28] with minor modifications. The following primers were used: *Hsp70* gene forward primer (F25) 5' -GGA CGC CAC GAT TKC T- 3' and reverse primer (R1310) 5' -CCT GGT TGT TCA GCC ACT C- 3'. For the ITS, the forward primer (LITSR): 5' -CTG GAT CAT TTT CCG ATG- 3' and reverse primer (LITSV): 5' -ACA CTC AGG TCT GTA AAC- 3' were used. The assays were conducted in 25µL reaction volume comprising: 12.5µL, 2X Mytaq Red Mix (Meridian Bioscience, USA), 0.5µL of each primer, 9.5µL of PCR grade water (Thermo Fisher Scientific, USA), and 2µL of the sample. The following conditions were used: For *Hsp70*: 94°C for 5 minutes, followed by 45 cycles of 94°C for 30 seconds, 61°C for 1 minute and 72°C for 3 minutes, followed by a final extension step at 72°C for 10 minutes. For ITS: 95°C for 2 minutes followed by 34 cycles of 95°C for 20 seconds, 53°C for 30 seconds, 72°C for 1 min, followed extension at 72°C for 6 min. The amplified products were visualized on a 2% agarose gel stained with 3.0µL GelRed (Biotium Inc., USA). Amplicons were cleaned using AmpureXP beads (Beckman Coulter, USA) following manufacturer's instructions, and then used to construct sequencing libraries with the Collibri ES DNA Library Prep Kit from Illumina (Thermo Fisher scientific, USA). The individual libraries for both *Hsp70* and ITS loci were normalized to 4nM concentration and then pooled separately. The pooled amplicon library was denatured and diluted to a final concentration of 8pM, then spiked with 5% PhiX (Illumina, USA) as a sequencing control and sequenced on the MiSeq platform, using v3 600 cycle kit; 2 × 300 bp paired reads (Illumina, CA, USA).

## Whole genome sequencing by target enrichment

In this proof-of-principle study, eight samples with high parasitemia as measured by genus-specific PCR (Ct values between 26–33) were selected for WGS. For SuSL-seq, two further criteria were added: the total DNA concentration and a measure of the % of *Leishmania* DNA in the sample: a minimum of 10 ng of total DNA and 0.006% of *Leishmania* DNA are required [23]. Total DNA was measured by qubit fluorometer (Thermo Fisher Scientific, USA) and 21–50ng used for library preparation. The quantification of *Leishmania* DNA in the library sample was performed by qPCR as described by Domagalska et al. [23]: *Leishmania* DNA in the samples ranged from 0.0064 to 0.1415%.

DNA fragmentation was performed enzymatically using the Agilent SureSelect Enzymatic Fragmentation Kit (Agilent, Santa Clara, USA). Libraries were prepared using the SureSelect XT HS Target Enrichment System for Illumina (Agilent, Santa Clara, USA). The probe design used in this study was based on a previously published SureSelect design [24]. Briefly, to generate the custom Agilent SureSelect (SuSL) capture array, repetitive and low-complexity regions of the reference genome (*Leishmania donovani* BPK282A1) were masked prior to probe selection. Candidate 120-mer probes were designed at 1x tiling density to minimize overlap, avoiding regions with ambiguous bases (Ns) and excessive sequence redundancy. To ensure high specificity, all probes were aligned using BLAST against the genome of Homo sapiens; any probes showing significant homology to these genomes were excluded to prevent off-target hybridization. The final design (Agilent SuSL design ID S3377046) comprised approximately 318,000 probes, collectively targeting ~29.99 Mb of the reference genome. Adaptor-ligated libraries were prepared using 11 cycles in pre-capture PCR. Libraries were hybridized with the custom probes at a dilution 1:10 and captured with Dynabeads MyOne Streptavidin T1 magnetic beads (Thermo Fisher Scientific, Waltham, USA). After washing steps, the DNA captured by streptavidin beads was amplified by PCR and purified with AMPure XP beads. The quantity and quality of the libraries were assessed on a TapeStation using High Sensitivity D1000 ScreenTape (Agilent Technologies, Santa Clara, USA). Sequencing was conducted on the Illumina NovaSeq platform using 2x150 bp paired reads at GenomeScan (Netherlands), for which 59,22–78,49 million raw reads were obtained.

## Analysis of data (targeted sequencing)

AAP3-qRT-PCR results for positive samples at genus level were tabulated as either negative or positive based on the set cycle threshold (Ct) values. For the *Hsp70* and ITS sequences, demultiplexed sequence reads from the Miseq were processed using the ngs_mapper (https://github.com/VDBWRAIR/ngs_mapper) pipeline which performed read trimming, reference mapping and consensus sequence generation. The consensus sequences were checked manually to resolve errors using Geneious prime 2022.2.1 (GraphPad LLC, MA, USA) and https://igv.org. Blastn v2.15.0 was used for homology analysis of the *Hsp70* (n = 86) and ITS (n = 79) sequences. Reference sequences obtained from GenBank (NC_018255.1 for *Hsp70* gene and NC_018254 for ITS region) were used to guide the homology analysis. To establish the phylogenetic relationship between Kenyan *Leishmania* spp and those from other regions, a comprehensive dataset of curated, annotated, and published *Leishmania* sequences were retrieved from NCBI GenBank (*Hsp70*, n = 67 and ITS, n = 65) (S2 Table) and aligned using the MAFFT v7.490; algorithm: E-INS-I in Geneious prime v.2022.2.1 (Auckland, New Zealand) plugin.

The *Hsp70* and ITS sequences were first analyzed separately and then concatenated to improve the phylogenetic resolution of individual genes. The concatenated sequences (n = 64) were aligned with previously published *Leishmania* sequences (n = 36) (S2 Table). The individual and concatenated genes were used to infer Maximum-likelihood (ML) trees using the best fitting nucleotide substitution model as determined by ModelFinder [29] within IQ-tree v2.2.26 (https://pubmed.ncbi.nlm.nih.gov/28481363/). Branch support was determined using both the ultrafast bootstrap approximation (UFboot) with 1,000 replicates and the SH-like approximate likelihood ratio test (SH-aLRT) also with 1,000 bootstrap replicates. Visualization and annotation of the resulting phylogenetic trees was done using Figtree version 1.4.2 (http://tree.bio.ed.ac.uk/software/Figtree). Taxa with unusually long branches had their consensus sequences re-evaluated for chimeras using UCHIME v4.2.40 against the QIIME eukaryotic database (release 2025-02-19), and the non-chimeric reads were used for consensus generation. The consensus sequence reads were then mapped to the *L. donovani* reference genome using the ngs_mapper pipeline. A Majority Rule Consensus was applied, meaning the base with the highest read support at each position was selected, ensuring the sequence represents the truly dominant allele. Positions not meeting sufficient coverage were masked with 'N's, guaranteeing a high confidence phylogenetic sequence.

## Analysis of data (whole genome sequencing)

For WGS by target enrichment, demultiplexed paired end reads from the NovaSeq6000 platform were assessed for quality using FastQC v0.14.1 (https://www.bioinformatics.babraham.ac.uk/projects/fastqc/) and FastQ Screen

(https://www.bioinformatics.babraham.ac.uk/projects/fastq_screen/). In addition to the genomes sequenced in the context of this work, we also added a set of representative sequences of the different clusters as defined in Franssen et al. [21] as available under the NCBI BioProject number PRJEB2600, PRJEB2724, PRJEB8947 and PRJEB2115 (S2 Table).

Reads were aligned to the *L. donovani* BPK282 reference genome [30] using BWA (v0.7.17) [31] with a seed length of 50. Only properly paired reads with a mapping quality >30 were retained using SAMtools [32]. Duplicate reads were removed using Picard (v2.22.4) with the RemoveDuplicates function. SNP calling followed the GATK (v4.1.4.1) best practices workflow: (i) HaplotypeCaller was used to generate GVCF files for each sample; (ii) individual GVCFs were merged using CombineGVCFs; (iii) genotyping was performed with GenotypeGVCFs; and (iv) SNP and INDEL filtering was applied using SelectVariants and VariantFiltration, with filtering criteria based on GATK best practices (QD > 2, QUAL > 30, SOR < 3, FS < 60, MQ > 40, MQRankSum > -12.5, ReadPosRankSum > -8.0) [33]. Variant regions associated with known drug resistance markers were extracted using BCFtools version 1.22 [34]. Impact of those mutations was assessed using the SnpEff software version 5.0c [35], and visualized with the heatmap package in R. Overlap of SNPs between two strains was determined using the vcfR [36] (v1.13.0) and adegenet (v2.1.8) [37] with visualization performed using pheatmap (v1.0.12).

For phylogenetic analysis, bi-allelic SNPs were extracted from VCF files using BCFtools version 1.22 and converted to Phylip format with the vcf2phylip.py script (https://github.com/edgardomortiz/vcf2phylip). Phylogenetic trees were inferred using RAxML, version 1.2. [38] under the GTR + G model, using *L. aethiopica* L100 as an outgroup. Tree visualization was performed with ggtree (v 3.14) [39]. To construct unrooted phylogenetic networks, bi-allelic SNPs were converted to FASTA format using the vcf2fasta.py script (https://github.com/FreBio/mytools/tree/master) and analyzed in SplitsTree4, version 6.4.13 using the NeighborNet algorithm [40].

For downstream population genomic analysis, SNP pruning for linkage disequilibrium (LD) was performed using PLINK 2.0 (--indep-pairwise 50 10 0.5) [41], with a threshold of $r^2 = 0.5$, which was selected to retain sufficient SNPs for population structure inference while minimizing the impact of tightly linked variants. This relatively permissive threshold was chosen because of the moderate LD decay observed in previous *Leishmania* genomic epidemiology studies [42–44] and to maintain representation across the genome despite local variation in recombination rates. ADMIXTURE v.1.3.0 [45] was used to assess ancestry in both WGS and SureSelect samples, excluding clonal individuals within each phylogenetic group. The number of populations (K) was evaluated from 4 to 11 using a five-fold cross-validation procedure. Population structure inferred by ADMIXTURE was visualized with ggplot2 in R.

For the exploration of possible genomic signatures of drug resistance, we selected 11 loci that were previously shown to be involved in *L. donovani* resistance to drugs used in clinical practice. Relevant regions [46–50] were subset using BCFtools, version 1.22 [34]. Only SNPs having an effect at the protein level (missense and non-sense mutations) were retained. Visualization of the heatmap was performed using the pheatmap function in R.

## Results

### Screening of Leishmania parasites

The demographics associated with the *Leishmania* samples is shown in S3 Table. Out of the 286 blood samples tested by AAP3-qRT-PCR, 128 (45%) were positive for *Leishmania* at the genus level. There was a 3:1 male:female infection ratio probably because of gender roles in this nomadic community. Men spend prolonged periods herding livestock where sand flies thrive. Additionally, seasonal migrations with livestock further expose men to diverse sand fly habitats, while women, who primarily engage in household activities, encounter less risk [13].

### Phylogenetic trees from individual genes

After quality control and assembly, we obtained consensus sequences of sizes 1,292 bp for *Hsp70* and 1,060 bp for ITS loci. Of the 128 samples positive for *Leishmania*, 86 generated usable sequences for *Hsp7*0 gene and 79 for the ITS

region (S4 Table). Homology analysis of the *Hsp70* (S2 Fig) and ITS (S3 Fig) sequences revealed monophyletic grouping of study samples that clustered with the *L. donovani* species complex obtained from the NCBI GenBank.

The closest relatives of the study sequences inferred from individual genes differed, with *Hsp70* (S2 Fig) showing phylogenetic proximity to countries north of Kenya (Sudan, Ethiopia) and ITS (S3 Fig) showing phylogenetic proximity to India, France, Spain, Morocco, Sri-Lanka, Sudan and Ethiopia. Given the different phylogenetic histories of *Hsp70* and ITS genes, we concatenated 64 study genomes against similar genomes available from the GenBank. In the concatenated tree, the study genomes clustered with the *L. donovani* complex in a group with *L. donovani* BPK282A1 (Fig 1). Like in S3 Fig., the study sample Kenya/GSA-164/2020 remained an outlier in the concatenated tree (Fig 1, shown in red star) and branched with the main *L. major* cluster, in a well-supported branch (bootstrap support of 63%). Because the *Hsp70* sequence for GSA-164 showed no substantial difference from other *L. donovani* strains in this study (S2 Fig), while the ITS sequences were highly divergent (S3 Fig), we verified that the observed divergence was not due to introduction of chimera.

### Direct Leishmania genome sequencing in clinical samples after target enrichment

Based on parasite load, eight blood samples with Ct values of <33 were selected (S5 Table) as pilots for direct targeted sequencing by SuSL-seq. Genome sequencing was successful for all the samples, with percentage of genome coverage > 5x ranging between 21.30% and 88.39% (average 67.97%). Summary of sequencing performances statistics can be found in S5 Table. Phylogenetic analysis of the eight genomes together with published genomes of *L. donovani* (including groups Ldon1–5, following classification by Franssen et al. [21], *L. infantum* and the three dermotropic species (*L. tropica, L. major* and *L. aethiopica*) [28] allowed us to identify the *Leishmania* clades in these 8 Kenyan samples. They clearly branched in the *L. donovani* s.s. cluster, separately from *L. infantum* and the dermotropic species (Fig 2). The eight Kenyan samples formed two distinct clades. Five of these samples clustered within the Ldon5 group, which includes isolates from Kenya and Ethiopia collected between 1954 (LRC-L53) and 2009 (AM560WTI) [21]; they were called K-Ldon5 (see Fig 2 red taxa). The remaining three samples clustered together close to three 'other Ldon' samples (LRV-L740 from Israel, GE, and LEM3472 from Sudan) in a distinct intermediate position in the rooted phylogenetic tree, between the Ldon3 group (Sudan/Ethiopia) and Ldon4 group (Iraq) (Fig 2); they were called K-Ldon3/Ldon4 (see Fig 2 purple taxa). In the reticulated network (S4 Fig), the three K-Ldon3/Ldon4 samples (purple taxa) branched separately from 'other Ldon' samples, suggesting an independent evolutionary origin. Interestingly, both K-Ldon5 and K-Ldon3/Ldon4 were present in both Garissa (Balambala and Kitui (Mwingi North) counties (see S1 Fig and S1 Table).

Given the intermediate phylogenetic position of the three Kenyan samples, we assessed if this genotype could have been generated by recombination between the previously reported *L. donovani* groups. Analysis of population structure was done in ADMIXTURE for *L. donovani* based on 61,186 genome-wide bi-allelic SNPs. After LD pruning, a total of 4050 SNPs remained and were used as input for ADMIXTURE analysis. We evaluated K = 2–20 using fivefold cross-validation. ADMIXTURE version 1.3.0 computes point estimates only, without standard errors, as noted in the log output. To assess model stability, we repeated each K value across 10 independent runs with different random seeds and obtained consistent ancestry proportions among replicates. The mean CV error values for each K are visualized including the standard error in S5 Fig, showing that CV error reached a minimum with K-values ranging between 4 and 9. Mixed ancestry (Ldon3 and Ldon4) is shown for isolates LRC-L740, LEM3472 and GE as reported earlier [21], at K = 5, 7 and 8 and the same pattern was observed in the three K-Ldon3/Ldon4 samples at K = 5 and 7 (Fig 3). In contrast, K-Ldon5 samples showed a single ancestral origin. Presence of mixed ancestry signatures in a sample could be reflecting events of genetic exchange as well as polyclonality (mixture of different genotypes). By plotting the alternative allele frequency along the 36 chromosomes of admixed sample GSA047, we found the signatures of genetic exchange, i.e., alternation of stretches of heterozygosity and homozygosity and average allele frequency of 50/50 in heterozygous stretches (see S6 Fig).

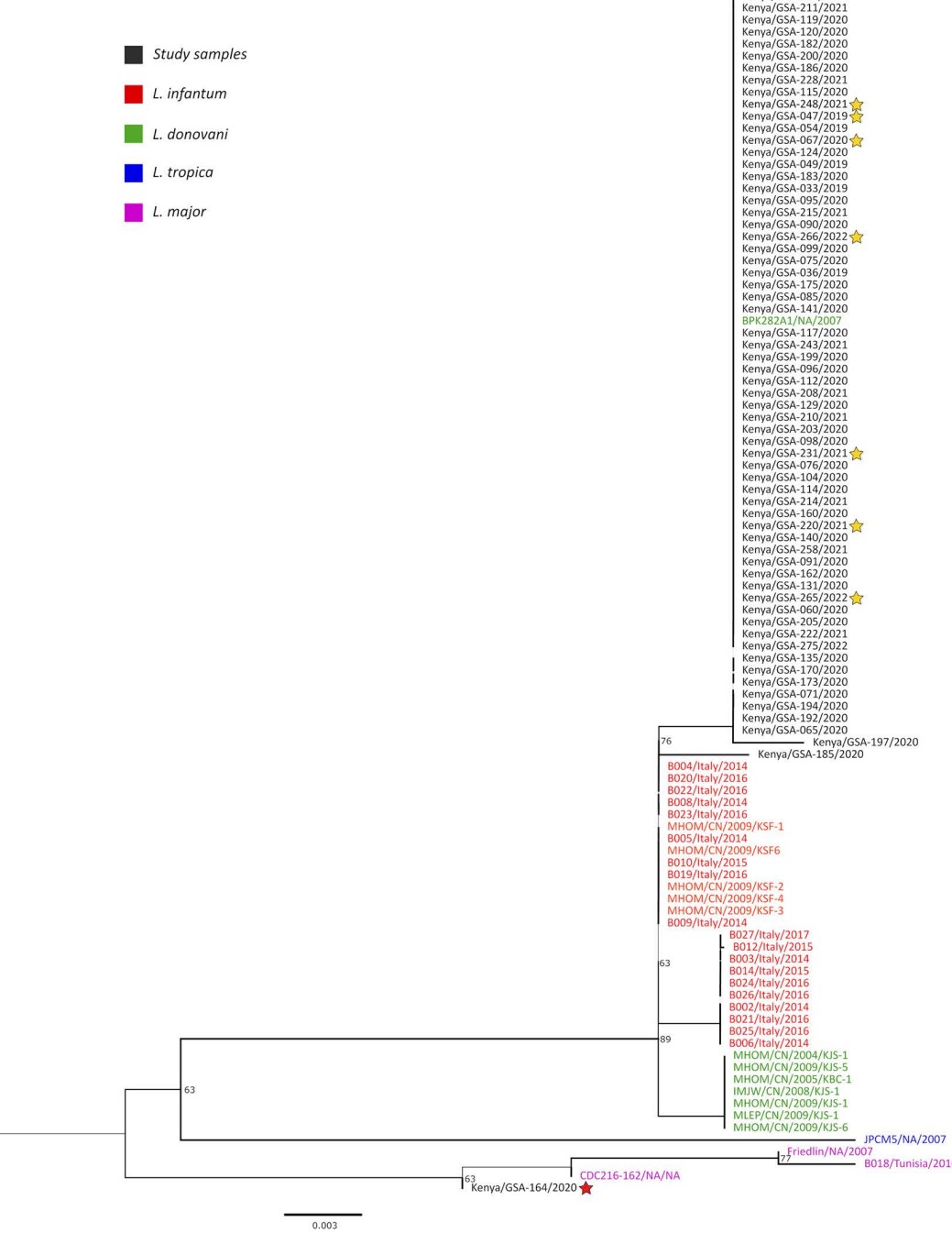

**Fig 1. Maximum-likelihood phylogenetic tree inferred from an alignment of concatenated study *Hsp70* and ITS sequences (black) and select *Leishmania* sequences retrieved from NCBI GenBank (colored text).** Red star shows an outlier study sample (Kenya/GSA-164/2020) that branched with the main *L. major* cluster. The yellow stars shows eight of the current study samples that were selected for targeted whole genome sequencing (see Fig 2). Branch support was estimated using SH-like likelihood ratio test and are indicated as numbers. The scale bar represents the number of substitutions per site.

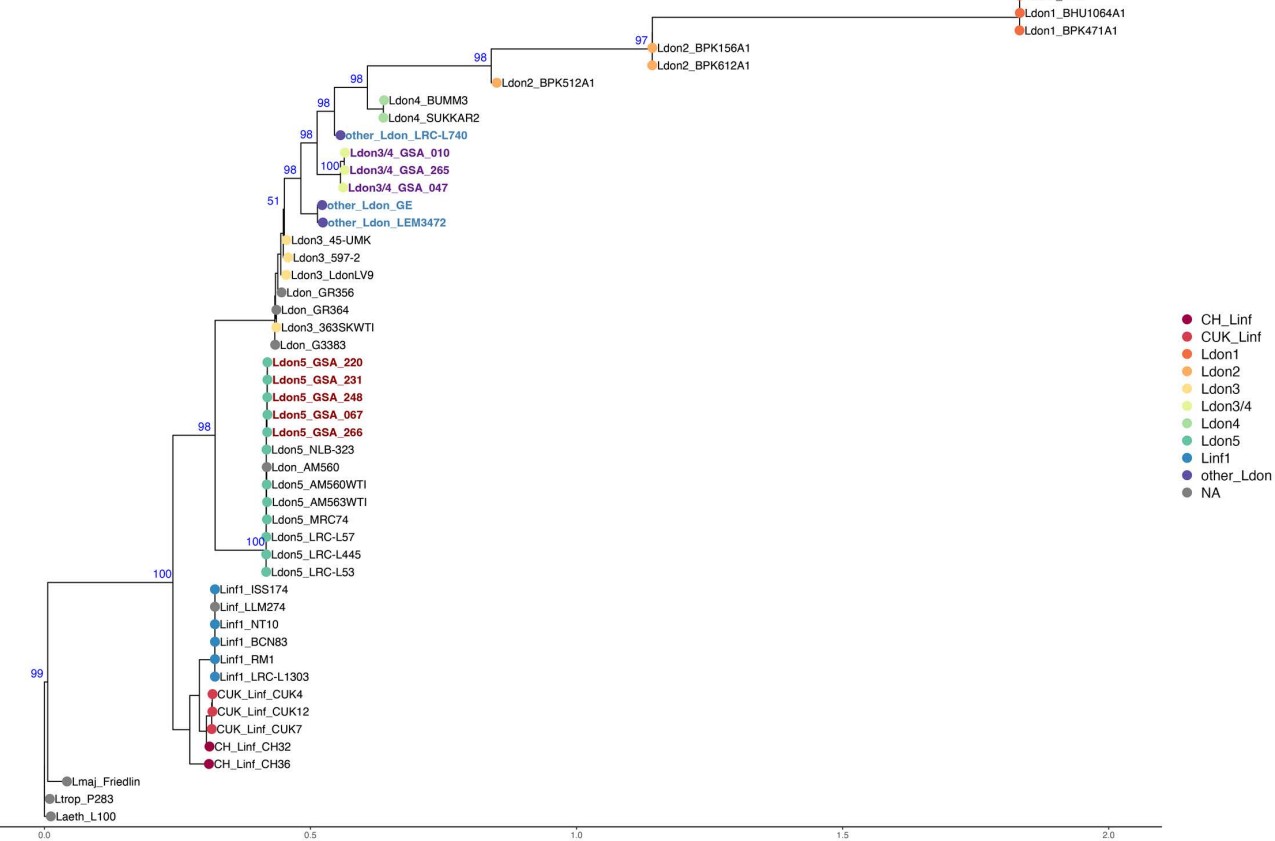

**Fig 2. Rooted phylogenetic tree based on genome-wide bi-allelic SNPs inferred using RAxML, with *L. aethiopica* L147 as the outgroup.** Most relevant bootstrap values are indicated in blue. The color code represents clustering as proposed by Franssen et al. [21] with samples from this study distributed across two clusters, highlighted in red and purple ("KenyaLeish"). The taxa in blue (three samples) indicate 'Other-Ldon' hybrid genotypes identified by Franssen et al. [21].

Finally, a scan of genomic signatures of resistance was undertaken on K-Ldon5, the three K-Ldon3/Ldon4 samples and a series of reference sequences. We first compiled a list of 11 genes reported to be involved in resistance to Antimony (Sb), Amphotericin B (AmB) and Miltefosine (MIL) [46–50] and analyzed SNPs having an effect at the protein level (missense and non-sense mutations). Overall, 46 SNPs were identified in the different genes that were analyzed. Of the 46 SNPs, 16 homozygous mutations related with resistance to Sb (MRPA), AmB (C5D) and MIL (MSL) were shared among the eight Kenyan samples and all Ldon5 reference genomes and LRC-L740 (S7 Fig). LiMT, the transporter of Miltefosine showed mutations in all Kenyan samples, but patterns differed between K-Ldon5 and K-Ldon3/Ldon4 samples: respectively, positions 486 and 183 and positions 248 and 495. In addition, one of the K-Ldon3/Ldon4 samples showed a mutation in AQP1 gene (see S7 Fig).

## Discussion

This study, for the first time in Kenya, used targeted and whole genome sequences from blood samples to characterize genomes of *Leishmania* parasites collected during the 2019–2022 VL outbreak in Garissa county. Multiple studies have documented outbreaks of VL in arid areas of North Eastern Kenya (Marsabit, Isiolo and Wajir counties) [51] and identified the influence of various environmental, ecological and social factors such as the presence of termite mounds around

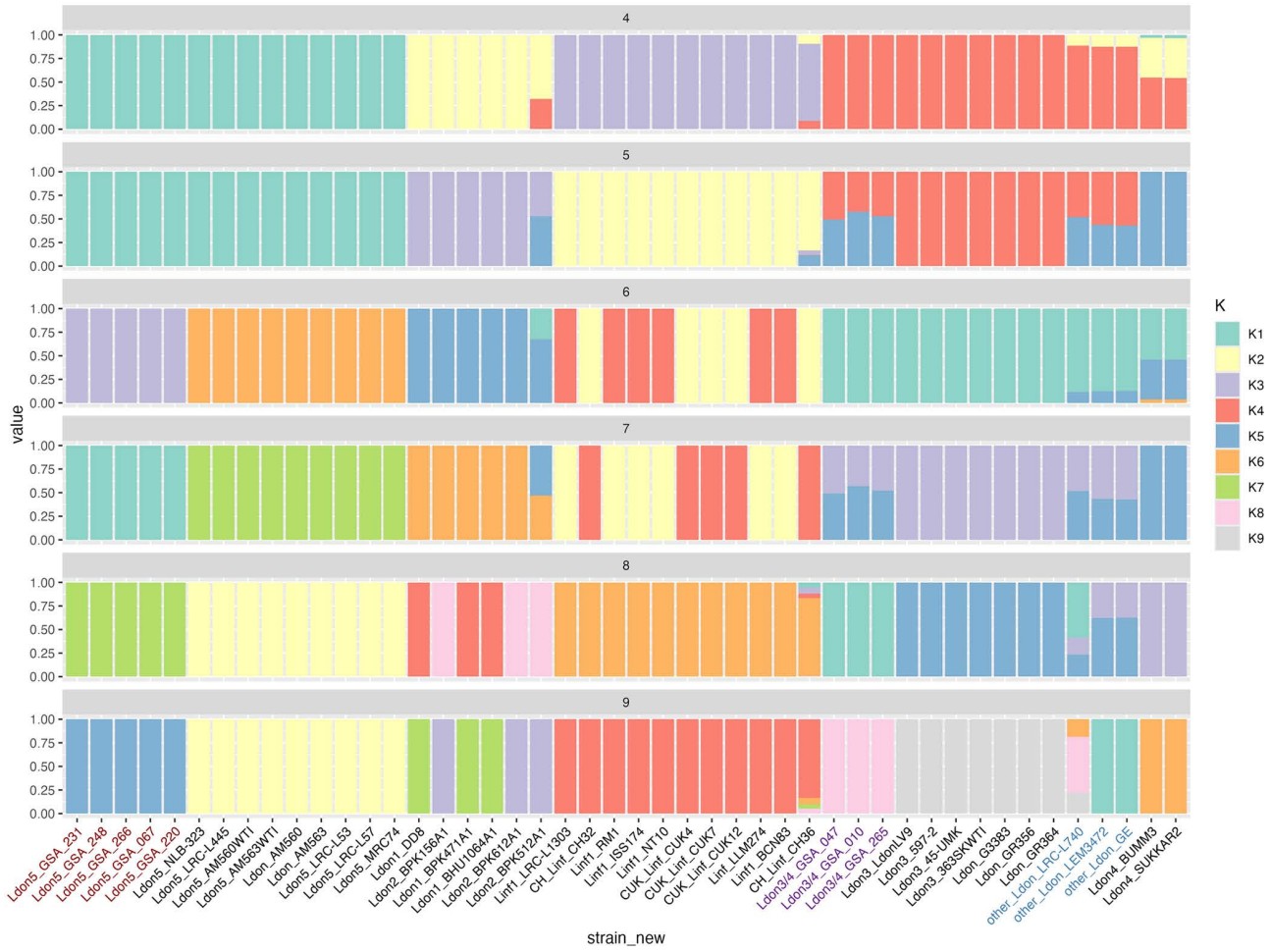

**Fig 3. Admixture analysis (K between 4 and 9, see S5 Fig); the samples in red and purple correspond to the 8 samples from Kenya (this study); each bar represents a sample, with colors corresponding to ancestry components.** Samples in blue indicate the genotypes with mixed ancestry reported by Franssen et al. [21].

homesteads, animal burrows, residing in rural sub-counties, and being male as being associated with higher VL infection rates [52].

In the current study, *Leishmania* genomes were first analyzed by targeting the *Hsp70* and ITS loci. We observed a slightly higher amplification success for *Hsp70* (n = 86) compared to ITS (n = 79), which we attribute to the intrinsic molecular properties of the targets and their respective primers. *Hsp70* is a conserved housekeeping gene that is essential for the parasite's survival and adaptation to various environmental stresses and has been widely used for phylogeny and taxonomy of *Leishmania* parasites [53,54]. It is highly conserved and thus has less chance of primer mismatches to target region. ITS region is extensively variable and is widely used for identification as well as phylogenetic analysis of *Leishmania* spp. [55]. Its high variability leads to better species discrimination, but at a cost of higher chance of primer mismatches that can reduce amplification yield.

Phylogenetically, all the 86 *Hsp70* sequences mapped to *L. donovani* complex from East African regions, indicating geographical kinship as has been suggested before [56]. Another interesting observation is the monophyletic nature of the study samples and those obtained from the NCBI GenBank, indicating the slow rate of evolution of the *L. donovani Hsp70*

gene. For instance, we observed a Kenyan sample collected in 1955 (S2 Fig., black star) clustering with our current study samples. We also noted that all samples from our study clustered closely with those from Ethiopia (S2 Fig).

Similar to *Hsp70*, the ITS phylogenetic tree was monophyletic (S3 Fig). But, unlike the *Hsp70*, the phylogenetic proximity of the study samples inferred using the ITS was India, France, Spain, Morocco, Sri-Lanka, Sudan and Ethiopia. This discrepancy arises from the fact that the *Hsp70* and ITS sequences deposited in the GenBank do not necessarily come from the same regions. Interestingly, one sample (Kenya/GSA_164/2020) separated distinctly from the other samples (S3 Fig, red star). By *Hsp70*, this sample did not resolve differently from the other study genomes (S2 Fig, red star). To improve the resolution and accuracy of the phylogenetic trees generated from individual genes, the *Hsp70* and ITS data were concatenated into single data set (Fig 1). In the concatenated tree, the study genomes clustered with the *L. donovani* complex. Like in the ITS data (S3 Fig), the study sample Kenya/GSA-164/2020 remained an outlier and branched with the main *L. major* cluster (Fig 1, red star), in a well-supported branch (bootstrap support of 63%). We contend that the observed phylogenetic distance of GSA-164 is not due to a mixed infection or a chimeric artifact. The single, unambiguous consensus sequence represents a highly divergent ITS genotype suggesting that GSA-164 may be a unique or novel lineage within *L. donovani* or carries a highly atypical set of ITS multicopy repeats.

We also combined target enrichment and WGS (SuSL-seq) for a pilot exploration of the *Leishmania* genome in 8 of the study samples. As shown in Fig 2, two distinct groups were identified: Group one comprising 5 study genomes that is related to *L. donovani* group 5 (Ldon5) and referred here as K-Ldon5 (K = Kenya). Group two comprised 3 samples of mixed ancestry, here referred to as K-Ldon3/Ldon4. The SuSL-seq protocol/application was developed to study parasite genomes directly from their natural environment and to avoid the strong selection biases associated with WGS of cultivated parasite isolates [23]. Indeed, in a previous study that used *Leishmania* samples from Nepal, it was demonstrated that the genome of parasites directly sequenced in bone marrow of VL patients was systematically different from the genome of in vitro derived isolates [23]. This led to recommendation that, studies needing to establish a link between parasite genome and clinical or epidemiological features should use parasites sequenced directly from clinical samples [22]. This led to the use of SuSL-seq for source tracing of *L. donovani* in emerging foci of VL in West of Nepal [24]. Direct sequencing of *Leishmania* parasites from blood has major advantages, among them facilitating sampling in remote places with minimal laboratory/medical facilities and allows discovery of new genotypes such as ISC11 described in the Nepal study [23] or the intermediate K-Ldon3/Ldon4 genotypes in the present study. It is currently unknown whether these seemingly new genotypes represent new genotypes or were under-reported because they were less adapted to culture conditions or represent true emergence of new genotypes in the study region. More direct sequencing studies involving past collections of *Leishmania* samples are needed in order to rule out sampling bias.

Given the phylogenetic proximity of the K-Ldon3/Ldon4 parasites and isolates LRV-L740, GE and LEM3472 in the phylogenetic tree (Fig 2), together with the admixture pattern similarity between these samples (Fig 3), we suggest to call this group Ldon6, hereby complementing the Ldon1-Ldon5 classification proposed by Franssen et al. However, this group is not monophyletic and it is a matter of convenience to class together parasites resulting from different admixture events between well differentiated populations Ldon3 and Ldon4, as shown by the reticulated network (S4 Fig) and previously hypothesized for LRV-L740, GE and LEM3472 [21]. Accordingly, analysis of additional samples is required to consider (re-)naming Ldon3/Ldon4 and in the meantime, it is more informative to keep the naming based on the combination of defined ancestry patterns.

The observation of mixed ancestry patterns could reflect recombination or polyclonality. The analysis of alternative allele frequencies along the 36 chromosomes of admixed samples allowed distinguishing the two hypotheses. In case of hybridization, stretches of heterozygosity should be alternating with stretches of homozygosity (intra-chromosomal recombination) and in heterozygous stretches, the frequency should always be 50/50 [57]. In case of polyclonal mixture, there should be no alternation of homozygous/heterozygous stretches and alternative allele frequency could deviate from 50/50. The hybridization pattern was observed for most chromosomes of K-Ldon3/Ldon4 samples. In the past, sexual

recombination in *Leishmania* was thought to be rare, but this topic was highly debated [58]. Recent studies show that hybrids are not that rare in natural populations; see for instance in *L. braziliensis* in Peru [59], *L. aethiopica* in Ethiopia [43], *L. tropica* in Morocco [42], *L. donovani* in India [60] and *L. infantum*/*L. donovani* in Italy [61]. We believe that this is due to (i) sampling which is more sympatric than before and (ii) the highest discriminatory power of WGS.

Our pilot study sheds some light on the genetic diversity within *L. donovani* complex, especially in East Africa where VL has become a hot spot. The observation of intra-specific genetic admixture confirms a trend increasingly reported in several species all over the world [59,62] and is likely explained by the high discriminatory power of WGS. In addition, in the selected genes reported to be involved in resistance to Antimony, Amphotericin B and Miltefosine [46–50], we identified a total of 46 SNPS in all the 8 Kenyan samples. Nevertheless, the presence of these drug resistance genes should be interpreted with care: at this stage, the genes represent genomic signatures and further phenotyping work is required to understand their functional implication in resistance. Further research on a larger sample size is needed to fully document the genomic heterogeneity of *L. donovani* in Kenya in past and current outbreaks (like Wajir and Pokot). Extension is also needed towards surrounding countries, in order to assess among others the epidemiological dynamics and the dispersal/spread of parasites across the continent and between continents (for instance Africa and the Indian sub-continent). Present studies and previous ones make a strong case for genomic surveillance studies using clinical samples so as to understand the changing landscape of *L. donovani*, which is essential to support control programs.

## Supporting information

**S1 Table. Demographics for the 8 blood samples selected for targeted whole genome sequencing.**
(DOCX)

**S2 Table. Genomes used in this study.**
(DOCX)

**S3 Table. The demographics of the patients who provided *Leishmania* samples for this study.**
(DOCX)

**S4 Table. Showing the relationship between *Leishmania spp* Ct values and amplicon success for both *Hsp70* and ITS markers.**
(XLSX)

**S5 Table. Summary of whole genome sequencing performances.**
(XLSX)

**S1 Fig. Map of Garissa and Kitui counties, Kenya, indicating the location of Garissa County Referral Hospital, where the visceral leishmaniasis samples were collected.** The dots indicate the reported origin of the 8 *Leishmania* samples (3 in Garissa and 5 in Kitui) analyzed by whole genome sequencing (WGS) by target enrichment (SuSL-seq). Samples clustered into two genomic groups: *L. donovani* group 5 (K-Ldon5: K for Kenya) similar to the previously reported *L. donovani* group 5. The other constituted a new genetic variant not reported previously (K- Ldon3/Ldon4). The map was generated using ArcGIS Pro 3.1. The base layer shapefile for Kenya's administrative boundaries was obtained from openAFRICA (https://www.open.africa/dataset/kenya-counties-shapefile) and is licensed under the Creative Commons Attribution 4.0 International (CC BY 4.0) license.
(TIF)

**S2 Fig. Maximum-likelihood phylogenetic tree showing the relationship between *Hsp70* gene sequences of the study samples (black) and selected *Leishmania* sequences retrieved from NCBI GenBank (colored fonts).** Black star shows a *L. donovani* sequence from Kenya that was deposited in GenBank in 1955 clustering with current study

samples. The yellow stars shows 8 of the current study samples that were selected for targeted whole genome sequencing (Fig 2). Branch support was estimated using SH-like likelihood ratio test and are indicated as numbers. The scale bar represents the number of substitutions per site.
(TIFF)

**S3 Fig. Maximum-likelihood phylogenetic tree showing the relationship between the ITS sequences of the study (black) and selected *Leishmania* sequences retrieved from NCBI GenBank (colored fonts).** Red star shows an outlier study sample (Kenya/GSA-164/2020), separating distinctly from others. The yellow stars shows 8 of the current study samples that were selected for targeted whole genome sequencing (Fig 2). Branch support was estimated using SH-like likelihood ratio test and are indicated as numbers. The scale bar represents the number of substitutions per site.
(TIF)

**S4 Fig. Phylogenetic network created based on all bi-allelic SNPs using the NeighborNet algorithm as implemented in SplitsTree4.** The samples in this study overlapping with cluster 5 *L. donovani* strains as defined by Franssen et al., [21] are shown in red. The three other strains not grouping with any of the known clusters are shown in a purple.
(PNG)

**S5 Fig. ADMIXTURE analysis.** Cross validation error as extracted from the ADMIXTURE log files for each tested value of the number of populations (K).
(PNG)

**S6 Fig. Allele frequency plot for 12 chromosomes in two samples, admixed (K- Ldon3/Ldon4 GSA047) and with single ancestry origin (K-Ldon5, GSA248).** The X-axis represents the position of the SNP in the chromosome, and the Y-axis represents the allele frequency of the SNP in the corresponding strain. Presence of mixed ancestry signatures in a sample could be reflecting events of genetic exchange as well as polyclonality (mixture of different genotypes). To distinguish the two hypotheses, we plotted the alternative allele frequencies along the 36 chromosomes of admixed samples. In case of polyclonal mixture, alternative allele frequency should be constant all along the chromosomes and it should deviate from 50/50. While in case of genetic exchange, stretches of heterozygosity should be alternating with stretches of homozygosity (intra-chromosomal recombination) and in heterozygous stretches, the frequency should always be 50/50. The latter was observed for most chromosomes of admixed sample, not for the sample with single ancestry origin. Noteworthy, the analysis could only be done with samples with similar and high genome coverage (see S5 Table): samples GSA47 and GSA248 show a coverage > 10% in 65,5 and 85.6% of the genome respectively (the two other admixed samples GSA010 and GSA265 showed only 4.4 and 26.5% of genome coverage).
(TIF)

**S7 Fig. Heatmap showing the distribution of single nucleotide polymorphisms (SNPs) in 11 genes reported to be associated with drug-resistant phenotypes.** The color scheme represents different SNP categories: blue indicates the absence of SNPs, orange indicates heterozygous SNPs, and red indicates homozygous SNPs. The naming convention for SNPs follows the format of the gene of interest, position in the genome, type of mutation, and its effect on the corresponding protein. The study samples are shown in red and purple text.
(PNG)

## Acknowledgments

We are grateful to the technical field assistants who supported the sample collection and logistics in Garissa County Referral Hospital. We also thank the patients whose samples are reported in this study. Special thanks to Rachael Githii for generating the map shown in S1 Fig.

## Author contributions

**Conceptualization:** Jean-Claude Dujardin, Malgorzata Anna Domagalska, John Waitumbi.

**Data curation:** Vane Kwamboka Omwenga, Gathii Kimita.

**Formal analysis:** Vane Kwamboka Omwenga, Gathii Kimita, Senne Heeren, Pieter Monsieurs.

**Funding acquisition:** John Waitumbi.

**Investigation:** Vane Kwamboka Omwenga, Damaris Matoke-Muhia, Cyrus Ayieko, Mohamed Hussein Ibrahim, Jean-Claude Dujardin, John Waitumbi.

**Methodology:** Vane Kwamboka Omwenga, Gathii Kimita, Damaris Matoke-Muhia, Clement Masakhwe, Mohamed Hussein Ibrahim, Senne Heeren, Jean-Claude Dujardin, Pieter Monsieurs, Malgorzata Anna Domagalska.

**Project administration:** Mohamed Hussein Ibrahim.

**Resources:** Jean-Claude Dujardin, John Waitumbi.

**Supervision:** Damaris Matoke-Muhia, Cyrus Ayieko, Clement Masakhwe, John Waitumbi.

**Visualization:** Vane Kwamboka Omwenga, Gathii Kimita.

**Writing – original draft:** Vane Kwamboka Omwenga.

**Writing – review & editing:** Vane Kwamboka Omwenga, Damaris Matoke-Muhia, Cyrus Ayieko, Senne Heeren, Jean-Claude Dujardin, Pieter Monsieurs, Malgorzata Anna Domagalska, John Waitumbi.

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
