## [Decision Letter · Decision Letter 0]

24 Sep 2025

Direct sequencing of Leishmania donovani from patients in Garissa County, Northern Kenya, reveals a newly emerging intra-specific hybrid genotype

Dear Dr. Waitumbi,

Thank you for submitting your manuscript to PLOS Neglected Tropical Diseases. After careful consideration, we feel that it has merit but does not fully meet PLOS Neglected Tropical Diseases's publication criteria as it currently stands. Therefore, we invite you to submit a revised version of the manuscript that addresses the points raised during the review process.

Please submit your revised manuscript within 60 days Nov 23 2025 11:59PM. If you will need more time than this to complete your revisions, please reply to this message or contact the journal office at plosntds@plos.org. Please include the following items when submitting your revised manuscript:

We look forward to receiving your revised manuscript.

Kind regards,

Susan Madison-Antenucci

Section Editor

Shaden Kamhawi

co-Editor-in-Chief

Paul Brindley

co-Editor-in-Chief

**Journal Requirements:**

1) Please provide an Author Summary. This should appear in your manuscript between the Abstract (if applicable) and the Introduction, and should be 150-200 words long. The aim should be to make your findings accessible to a wide audience that includes both scientists and non-scientists. Sample summaries can be found on our website under Submission Guidelines:

2) Some material included in your submission may be copyrighted. According to PLOSu2019s copyright policy, authors who use figures or other material (e.g., graphics, clipart, maps) from another author or copyright holder must demonstrate or obtain permission to publish this material under the Creative Commons Attribution 4.0 International (CC BY 4.0) License used by PLOS journals. Please closely review the details of PLOSu2019s copyright requirements here: PLOS Licenses and Copyright. If you need to request permissions from a copyright holder, you may use PLOS's Copyright Content Permission form.

Potential Copyright Issues:

- Figure S1. Please (a) provide a direct link to the base layer of the map (i.e., the country or region border shape) and ensure this is also included in the figure legend; and (b) provide a link to the terms of use / license information for the base layer image or shapefile. We cannot publish proprietary or copyrighted maps (e.g. Google Maps, Mapquest) and the terms of use for your map base layer must be compatible with our CC BY 4.0 license.

3) Please ensure that the funders and grant numbers match between the Financial Disclosure field and the Funding Information tab in your submission form. Note that the funders must be provided in the same order in both places as well.

**Reviewers' Comments:**

Reviewer's Responses to Questions

**Key Review Criteria Required for Acceptance?**

**Methods:**

-Are the objectives of the study clearly articulated with a clear testable hypothesis stated?

-Is the study design appropriate to address the stated objectives?

-Is the population clearly described and appropriate for the hypothesis being tested?

-Is the sample size sufficient to ensure adequate power to address the hypothesis being tested?

-Were correct statistical analysis used to support conclusions?

-Are there concerns about ethical or regulatory requirements being met?

Reviewer #1: My first and most important concern is the ethical context of the work. Samples have been collected from Kenyan individuals. Yet, there has been no ethical review done and/or obtained from a Kenyan institute. An USA-based institute deemed that this was not needed, but I do not think that this is a leading opinion.

Secondly for the WGS approach (8 samples) approval was obtained from ITM (Belgium), but this does not concern the use of the other specimen. I really think that an opinion from a Kenyan based institute should have been sought. This must be addressed by the authors.

The authors should also explain why only 8 samples were selected for WGS.. Is this number sufficient to support the conclusions. The rationale should be presented in M&M section, its impact on the conclusions in the discussion

Reviewer #2: Study design, methods, statistical analysis is correctly used. For ethics, in this study, it is stated that the work was conducted as part of the public health response during the outbreak. I am wondering, in addition to the routine diagnosis, whether research use of the samples, such as WGS is also included in this permission? If the Kenyan ethical review committee has determined that such approval is not necessary, it would be advisable to clearly state this to avoid any potential misunderstandings.

Reviewer #3: Methods presented are ok, but some missing key experimental desing are needed in the current state

**Results**

-Does the analysis presented match the analysis plan?

-Are the results clearly and completely presented?

-Are the figures (Tables, Images) of sufficient quality for clarity?

Reviewer #1: Table 1 is not very informative, possibly it could be removed. If maintained more demographic information must be added. Also the heading should specify that it concerns a specific locality.

Reviewer #2: Results align withs the methods and analysis plan, results are clearly and completely presented. The figures are well-designed and thoughtfully constructed, which helps to simplify the interpretation of the results. This effectively conveys the clarity of the findings.

Reviewer #3: Can be improved

**Conclusions**

-Are the conclusions supported by the data presented?

-Are the limitations of analysis clearly described?

-Do the authors discuss how these data can be helpful to advance our understanding of the topic under study?

-Is public health relevance addressed?

Reviewer #1: Samples have been collected about 5 years ago and from a specific region. Nowadays Kenya is confronted with other major outbreaks, Wajir, Pokot. How do the authors think that the results of their work translate to these recent outbreaks?

Reviewer #2: See general comment

Reviewer #3: Needs revision

**Editorial and Data Presentation Modifications?**

Reviewer #1: See my comment above on ethics. This must be addressed

Reviewer #2: Line 305

"For, instance" should be corrected to "For instance".

Figure 3 – Color and Labeling

The yellow area missing strain names. Also, the orange representing heterozygous SNPs seems to be yellow, which could be confusing.

Supplementary Table S3

It would be helpful to include the percentage of parasite reads for each sample. Including Ct values as well may allow readers to assess whether enrichment efficiency correlates with the initial parasite load or is influenced by other factors such as sample DNA quality.

Figure 3

Consider marking GSA-164 with a red star to improve readability

Reviewer #3: (No Response)

**Summary and General Comments**

Reviewer #1: This is a well written and presented paper. It provides a significant insight in the genetic diversity of Leishmania parasites relevant for Kenya, but it is of interest beyond this region.

Reviewer #2: The authors conducted a detailed genetic analysis of Leishmania donovani outbreak strains in Kenya using a target enrichment approach. This method enabled high-resolution phylogenetic analysis that would not have been possible through conventional single gene sequence analysis. Their results reveal that the outbreak did not originate from a single clonal lineage but instead consisted of genetically at least two distinct parasite populations. Furthermore, these populations appear to constitute a novel group that does not cluster with any previously known groups, and the authors propose to designate this as a new clade, “Ld6”. Importantly, genome-wide SNPs obtained directly from clinical samples also suggest potential drug resistance. These findings provide significant implications for treatment strategies and future outbreak responses in Kenya. The study is well-conducted, and the manuscript is clearly written and easy to follow.

Line 222: GSA-164

The authors discuss GSA-164, which showed divergent behavior in ITS and HSP-based phylogenetic analyses, possibly due to low Ct values. While HSP sequences showed no substantial difference from other Ld strains in this study, ITS sequences appear to be highly divergent. The phylogenetic distance observed in the ITS sequence is unusually large, making interpretation difficult. According to the methods, the ITS amplicon was sequenced using Illumina MiSeq following library preparation, which likely included enzymatic fragmentation and subsequent clustering by bioinformatics processing.The authors should briefly describe how the consensus sequence for this sample was generated. Was a single, unambiguous ITS consensus sequence obtained for this sample? Could there be a minor component in the original sample—such as L. major or another heterologous population—that introduced potential chimeric sequences either through PCR template switching or during downstream analysis?

Ct Threshold

The authors used a Ct cutoff of 33 for sample selection for SureSelect, with successful results. Is there a rationale or empirical basis for selecting this Ct threshold? A brief explanation should also be provided regarding why whole-genome sequencing could not be performed for GSA-164.

Discussion

The authors might consider elaborating on the ecological background that could explain the occurrence of hybridization within the Ld lineage. Assuming that L. donovani is anthroponotic with no animal reservoir, the opportunities for recombination in nature would be quite limited, likely requiring co-infection of sandflies that fed on multiple human hosts. This would imply that such events are rare and not frequently occurring.

Reviewer #3: This is an interesting and timely epidemiological/genomic study. To my knowledge, it is the first application in Africa of targeted enrichment directly from patient blood to sequence Leishmania genomes, avoiding culture bias and enabling insights into population structure and potential drug-resistance signatures. These contributions are important for public health surveillance. I have several substantive comments that, if addressed, will strengthen the paper’s impact and technical clarity.

Major comments

- Frame the study around the Ld1–Ld5 (Franssen) subdivision and why it matters

The Introduction nicely summarizes VL epidemiology, but it doesn’t prepare the reader for the paper’s central population-genetic result which is placement of Kenyan isolates relative to the Ld group framework and the proposal of an additional “Ld6-like” set (Results/Discussion, including Fig. 2 and related text). Please expand the Introduction to (i) explain why Ld groupings (Franssen et al.) are a useful organizing principle; (ii) set expectations that multi-locus or whole-genome SNP phylogenies integrate signals across many genes (more pangenome-like than single-marker trees), and (iii) preview why a putative new cluster would be meaningful beyond relabeling. This context will make later sections more interpretable.

- Methods for targeted enrichment: describe probe design and report capture performance

The Methods specify a custom Agilent SureSelect design (SuSL design ID S3377046; ~318k 120-mer probes covering ~29.99 Mbp) and NovaSeq 2×150 bp sequencing, which is excellent. Please clarify whether this probe set is newly designed here or previously published and, in either case, cite the design source and criteria (tiling density, target regions, species coverage). Beyond read counts, the manuscript should report standard capture metrics per sample: on-target rate, mean depth, duplication rate (post-dedup), and breadth of coverage. Provide per-chromosome (or genome-wide) coverage plots, and relate capture success to Ct values (you selected Ct 26–33; Methods). This QC is essential for judging how robustly SuSL-seq worked on blood and how that impacted downstream analyses (e.g., allele-frequency plots rely on adequate coverage).

- Case definition and consistency (“suspected” vs. “confirmed”)

The Abstract and some Methods describe samples from “patients suspected of having VL,” yet “Study site and design” states “patients with visceral leishmaniasis” while only 128/286 (45%) were Leishmania-positive by AAP3 qPCR (Results, Table 1). Please standardize to “patients suspected of VL” in the body text for accuracy and consistency with your own numbers.

- Amplicon marker success and rationale

You recover 86 Hsp70 and 79 ITS sequences (Results). Please comment on why yields differ (e.g., DNA quality, Ct, primer mismatch) and provide a simple success-rate summary by Ct strata. A small supplemental table or figure relating Ct to amplicon success would make the targeted marker section more informative. In summary: Add a supplemental CT table for all samples (amplicon and WGS-selected), plus a coverage summary table for the eight WGS samples: raw reads, % on-target, mean depth, % genome, % duplicates. Include a coverage-vs-Ct scatterplot to illustrate performance boundaries of SuSL-seq in blood.

- Population structure and the “new cluster”

Your ADMIXTURE and phylogenomic analyses suggest three Kenyan samples sit between Ld3 (Sudan/Ethiopia) and Ld4 (Iraq), with admixture like previously noted hybrids. Given the small WGS sample size and the authors’ own acknowledgement that the group is not monophyletic (Discussion), I recommend softening the claim from “new cluster Ld6” to “an admixed genotype” (or similar) unless additional sampling or tests are added to support a stable, reproducible group. The Discussion should also note that recombination and polyclonality can both produce mixed ancestry patterns; your allele-frequency analysis is a good step build on it with clearer thresholds/criteria.

- Drug-resistance SNPs: present as genomic signals unless functionally validated

The heatmap of SNPs in 11 AMR-implicated genes is eye-catching, but without phenotypic or epidemiological correlation it should be framed as putative genomic signatures. If phenotypic validation is not feasible now, consider moving the heatmap and gene-by-gene calls to the Supplement and dialing back claims in the main text, or add stronger caveats in Results/Discussion.

- Show the link between LD pruning, SNP counts, and ADMIXTURE model choice

You prune with PLINK and evaluate K=4–11 by CV error. Please (i) justify the LD threshold (r²=0.5 is fairly permissive), (ii) report the final SNP count after pruning, and (iii) include the CV error values with error bars or replicate runs to show stability. This will help readers evaluate the K choice and ancestry patterns.

Minor comments

Line ~93 (Study site/design): Replace “patients with visceral leishmaniasis” with “patients suspected of visceral leishmaniasis,” consistent with your Abstract and 45% qPCR positivity.

Lines ~103–105 (AAP3 qPCR mix): “Meridian Bioscience, USA” is duplicated—please remove the duplicate and keep the one with web link once.

Line ~125 (MiSeq): Specify “MiSeq v3 600-cycle kit (2×300 bp)” explicitly to be consistent with the 2×150 bp NovaSeq run later (line ~141).

Line ~154 (MAFFT): Add the MAFFT version (and algorithm, e.g., L-INS-i/G-INS-i) used via the Geneious plugin.

Lines ~172–183 (Mapping & GATK): You list GATK filters but no minimum depth; please state a DP threshold (e.g., DP ≥10) used in filtering and in downstream analyses.

Line ~181: Add SnpEff version.

Line ~186: Add RAxML/RAxML-NG version

Line ~189: Add SplitsTree4 version.

Line ~192: Add ADMIXTURE version and random seed(s) for replication.

Line ~198: Add BCFtools version.

Figures: Improve resolution/legibility and enforce consistent nomenclature. For example, use a single convention (e.g., Ld5) across text and figures instead of mixing Ld5/Ldon5. Also ensure sample labels match between Fig. 2 and Fig. 3 and that axis labels and legends are readable at journal scale.

PLOS authors have the option to publish the peer review history of their article (what does this mean? ). If published, this will include your full peer review and any attached files.

**Do you want your identity to be public for this peer review?** For information about this choice, including consent withdrawal, please see our Privacy Policy .

Reviewer #1: No

Reviewer #2: No

Reviewer #3: No

**Figure resubmission:**
---

## [Decision Letter · Decision Letter 1]

15 Jan 2026

Dear Dr. Waitumbi,

We are pleased to inform you that your manuscript 'Direct sequencing of Leishmania donovani from patients in Garissa County, Northern Kenya, reveals a newly emerging intra-specific hybrid genotype' has been provisionally accepted for publication in PLOS Neglected Tropical Diseases.

Best regards,

Susan Madison-Antenucci

Section Editor

Shaden Kamhawi

co-Editor-in-Chief

Paul Brindley

co-Editor-in-Chief

Reviewer's Responses to Questions

**Key Review Criteria Required for Acceptance?**

**Methods**

-Are the objectives of the study clearly articulated with a clear testable hypothesis stated?

-Is the study design appropriate to address the stated objectives?

-Is the population clearly described and appropriate for the hypothesis being tested?

-Is the sample size sufficient to ensure adequate power to address the hypothesis being tested?

-Were correct statistical analysis used to support conclusions?

-Are there concerns about ethical or regulatory requirements being met?

Reviewer #1: (No Response)

Reviewer #2: (No Response)

**Results**

-Does the analysis presented match the analysis plan?

-Are the results clearly and completely presented?

-Are the figures (Tables, Images) of sufficient quality for clarity?

Reviewer #1: (No Response)

Reviewer #2: (No Response)

**Conclusions**

-Are the conclusions supported by the data presented?

-Are the limitations of analysis clearly described?

-Do the authors discuss how these data can be helpful to advance our understanding of the topic under study?

-Is public health relevance addressed?

Reviewer #1: (No Response)

Reviewer #2: (No Response)

**Editorial and Data Presentation Modifications?**

Reviewer #1: (No Response)

Reviewer #2: (No Response)

**Summary and General Comments**

Reviewer #1: based on the revision letter, I think that the authors have mostly addressed my main concerns, in particular those with respect to ethical review. Unfortunately, the authors did not provide a revised manuscript with track changes, so it is difficult to follow which changes have been actually made. I also noted that betwwen lines 252 and 253 there is an empty table that should be taken out.

Reviewer #2: (No Response)

PLOS authors have the option to publish the peer review history of their article (what does this mean? ). If published, this will include your full peer review and any attached files.

**Do you want your identity to be public for this peer review?** For information about this choice, including consent withdrawal, please see our Privacy Policy .

Reviewer #1: No

Reviewer #2: No

---

## [Editor Report · Acceptance letter]

Dear Dr. Waitumbi,

We are delighted to inform you that your manuscript, "Direct sequencing of Leishmania donovani from patients in Garissa County, Northern Kenya, reveals a newly emerging intra-specific hybrid genotype," has been formally accepted for publication in PLOS Neglected Tropical Diseases.

Best regards,

Shaden Kamhawi

co-Editor-in-Chief

Paul Brindley

co-Editor-in-Chief
